

# Analysis of novel therapeutic targets and construction of a prognostic model for hepatocellular carcinoma

Pengyuan Lei[1,2,*], Wenfeng Li[2,*], Dafei Xie[2], Hua Guan[1,2], Xin Huang[2] and Bo Huang[1]

[1] College of Public Health of University of South China, Hengyang, China
[2] Beijing Key Laboratory for Radiobiology, Department of Radiation Biology, Beijing Institute of Radiation Medicine, Beijing, China
[*] These authors contributed equally to this work.

## ABSTRACT

**Background.** The prognosis of patients with hepatocellular carcinoma (HCC) remains suboptimal due to limited biomarkers. Although ferroptosis and cuproptosis have emerged as promising therapeutic targets, their prognostic significance in HCC remains unclear.

**Methods.** This study analyzed the expression of ferroptosis- and cuproptosis-related genes associated with survival in HCC, utilizing datasets from The Cancer Genome Atlas and the Gene Expression Omnibus. The impact of clinical factors on patient prognosis was also analyzed. The key findings were validated using the Human Protein Atlas database, quantitative real-time PCR, and Western blot (WB) analyses.

**Results.** Prognostic modeling identified six ferroptosis-related genes (SLC1A5, SLC7A11, CBS, GABARAPL1, FLT3, and MT3) and three cuproptosis-related genes (ADM, CDKN2A, and GLS) significantly associated with HCC prognosis. A robust risk assessment model was developed with strong predictive power. The inflammatory cell apoptotic process is an immune-related pathway that is commonly enriched by ferroptosis and cuproptosis genes. Immune cell profiling using CIBERSORT revealed significant differences in macrophages, naive B cells, mast cells, monocytes, memory CD4+ T cells, and CD8+ T cells.

**Conclusions.** This study, for the first time, integrated multi-omics data and experimental verification to establish a prognostic model of HCC based on the ferroptosis gene. It can be used to dynamically monitor HCC using blood tests and provide a new target for personalized immunotherapy.

Corresponding authors
Xin Huang,
huangxin.bio@foxmail.com
Bo Huang, huangbo0930@163.com

## INTRODUCTION

According to the latest *Global Cancer Statistics Report*, 665,000 new cases of liver cancer occur every year, leading to 757,948 deaths, making it the sixth most common malignant tumor and the third predominant cause of tumor-related death in the world (*Anonymous, 2024*; *Bray et al., 2024*; *Ding et al., 2021*). *The Key Points of Global Cancer Statistics 2022*

reported that liver cancer ranks as the fourth most prevalent cancer in China and second in cancer-associated mortality rates (*Liu et al., 2024b*; *Qi et al., 2023*; *Yu et al., 2023*). Hepatocellular carcinoma (HCC) is a particularly fatal primary liver cancer with an extremely high fatality rate and a relatively low 5-year survival rate of only 22% (*Siegel, Giaquinto & Jemal, 2024*). Its poor prognosis is linked to easy metastasis, late diagnosis, and a high rate of relapse (*Feng et al., 2022*). This severe clinical situation urgently requires more effective prognostic biomarkers, and the core role of abnormal cell death pathways in cancer progression provides a new perspective for this approach. Ferroptosis and cuproptosis, as emerging cell death patterns, have received extensive attention due to their unique molecular mechanisms and potential correlation with cancer biology (*Du, Zeng & Deng, 2023*; *Wang et al., 2023a*).

Ferroptosis is characterized by lipid peroxidation and the excessive accumulation of reactive oxygen species (*Benedetti, Jézéquel & Orlandi, 1988*; *Jiang, Stockwell & Conrad, 2021*). Numerous studies suggest that targeting ferroptosis is crucial in eliminating tumor cells and hindering their proliferation, particularly in HCC (*Wang et al., 2023b*). Sorafenib is the preferred drug for treating advanced liver cancer, which hampers the survival, growth, and movement of HCC cells in a dose-dependent manner. The drug alters the mitochondrial structure, resulting in diminished oxidative phosphorylation, a drop in mitochondrial membrane potential, and decreased adenosine triphosphate (ATP) production, ultimately causing cell death through ferroptosis (*Fu et al., 2023*; *Pinyol et al., 2019*; *Stockwell, 2022*). Blocking the p62-Keap1-NRF2 pathway, which activates MT1, was reported to improve sorafenib resistance and induce ferroptosis (*Liu & Li, 2022*; *Pinyol et al., 2019*). *Chen et al. (2022)* reported that SOCS2 expression promoted ferroptosis by facilitating the ubiquitination degradation of SLC7A11, which may increase the efficiency of HCC radiotherapy and improve patient prognosis. Triggering ferroptosis suppresses the proliferation of HCC cells, thereby reversing tumorigenesis, improving the efficacy of immunotherapy, and enhancing the anti-tumor immune response (*Liu et al., 2024b*). Further studies of ferroptosis in HCC may identify prognostically relevant ferroptosis genes.

Cuproptosis refers to the process whereby excessive copper promotes the aggregation of lipidized proteins and the instability of Fe-S cluster proteins, which leads to protein toxicity and, ultimately, cell death (*Tsvetkov et al., 2022*). It is closely associated with mitochondrial dysfunction, as excess copper ions damage the mitochondrial membrane structure, leading to the loss of mitochondrial function and the accumulation of intracellular reactive oxygen species (ROS), which, in turn, triggers cell death (*Xing et al., 2023*). Research reports suggest that targeting copper-induced cell death *via* the tricarboxylic acid (TCA) cycle is a promising therapeutic strategy. In HCC, the loss of ARID1A causes a shift in cellular glucose metabolism from aerobic glycolysis to reliance on the TCA cycle and oxidative phosphorylation (*Xing et al., 2023*).

Although ferroptosis shows pro-apoptotic effects in hepatoma cell lines, and copper death is associated with metabolic abnormalities in hepatoma, the systematic analysis of the two in clinical prognosis and their interaction with the immune microenvironment remain unclear.

This research primarily aimed to investigate ferroptosis and cuproptosis genes associated with the prognosis of HCC and establish a predictive model for HCC patients by searching and mining an HCC RNA-sequencing (RNA-seq) public dataset and related clinical information in The Cancer Genome Atlas (TCGA). This study lays the foundation for in-depth research on ferroptosis in HCC and discovers novel treatment targets for HCC.

## MATERIALS & METHODS

### Data collection and processing

The genome sequencing data of patients with HCC (RNA-seq) and the related clinical information were obtained from TCGA (https://www.cancer.gov/ccg/research/genome-sequencing/tcga). The dataset included 374 samples from patients with HCC and 50 healthy samples. We excluded samples lacking clinical data or with an overall survival (OS) of <30 days. Thus, 358 samples were included for subsequent analysis. We also downloaded external data from the Gene Expression Omnibus (GEO) (https://www.ncbi.nlm.nih.gov/geo/) and HCCDB v2.0 (http://lifeome.net/database/hccdb2) for validation and obtained genes from the FreeDb website (http://www.zhounan.org/ferrdb/).

### Analysis of genes with varied expression levels

The DESeq2 software in R (version 4.3.1; *R Core Team, 2023*) was utilized to analyze differentially expressed genes (DEGs) between patients with HCC and healthy individuals. The criteria for selecting DEGs were $|log2FoldChange|>1$ and $p < 0.05$. The original RNA-seq count was normalized using the median ratio method of DESeq2. Genes with counts less than 10 in the sample were excluded.

### Identification and analysis of survival-related novel death modality genes affecting HCC prognosis

A univariate Cox regression analysis was used for survival-related novel death modality-related DEGs using the "survival" package in R software. LASSO-Cox regression was implemented *via* "glmnet" with 10-fold cross-validation. The optimal λ (lambda = 0.032) maximized partial likelihood deviance reduction while retaining ≤10 predictors to prevent overfitting. Subsequently, we employed multivariate Cox regression analysis to identify risk genes potentially affecting HCC prognosis. Risk scores (RSs) were calculated using a specific formula: $RS = \sum \beta_i \times gene_i$ expression. High and low RS groups were categorized based on the median RS. Survival analysis utilized Kaplan–Meier graphs, and time-dependent receiver operating characteristic (ROC) curves over 3 and 5 years were computed to assess the precision of the risk assessment models.

### Clinical factor correlation analysis and establishment of a nomograph

Univariate and multivariate Cox regressions were used to analyze the relationship between clinical factors and RS (gender, age, clinical stage, et cetera). Using R's "rms" package, we plotted a nomo-diagram to predict the survival of patients with HCC at 1, 2, and 3 years.

## Gene expression profile interaction analysis

The gene expression profile interaction analysis (GEPIA) website (http://gepia2.cancer-pku.cn#analysis) was employed to assess the expression levels of ferroptosis-related mRNAs in HCC ($n = 404$) and normal samples from the TCGA.

## Protein expression validation

Immunohistochemical staining images of gene protein expression in HCC and normal samples were extracted from the Human Protein Atlas (HPA) database (https://www.proteinatlas.org).

## Human HCC cell culture

The THLE-2 normal human liver cell line was cultured in THLE-2-specific medium. MHCC97H, a hepatocellular carcinoma cell line, was cultured in normal Dulbecco's modified Eagle's medium (DMEM; HyClone, Logan, UT, USA), both at 37 °C and 5% $CO_2$. Both cell lines were purchased from the American Type Culture Collection (Manassas, VA, USA).

## Quantitative real-time PCR

Total RNA was extracted using TRIzol reagent (Invitrogen, Carlsbad, CA, USA) according to the manufacturer's protocol. One microgram of RNA was reverse-transcribed using HiScript®III RT SuperMix for qPCR (+ gDNA wiper) (R323-01) (Vazyme Biotech, San Diego, CA, USA). qPCR was performed on diluted cDNA using Taq Pro Universal SYBR qPCR Master Mix (Q712-02) (Vazyme Biotech).

The primers used were as follows:
CBS forward: 5′- GGCCAAGTGTGAGTTCTTCAA-3′ and
CBS reverse: 5′-GGCTCGATAATCGTGTCCCC-3′

## Western blotting analysis

Cultured hepatocytes were lysed with lysate (89901; Thermo Fisher Scientific) for 15–20 min on ice, sonicated for 1 min, centrifuged at 12,000 rpm for 15min at 4 °C, and the supernatant was collected. Protein quantification was performed using a Thermo NanoDrop 2000 Ultramicro ultraviolet spectrophotometer. A 72.5 uL volume of 5× sodium dodecyl sulfate (SDS) buffer was added and boiled in boiling water at 100 °C for 15 min under open cover. Proteins were separated by electrophoresis at 80 V for 30 min and 120 V for about 1 h on a 12% SDS separating gel and a 5% concentrating gel. The sandwich structure: thick filter paper → nitrocellulose (NC) film → SDS-polyacrylamide gel electrophoresis glue → thick filter paper clip was put into the transfer tank of an ice bath. After the film was turned at a constant pressure of 100 V for 1 h, the membrane was blocked with 5% skim milk, and the membrane was closed at room temperature by shaking at a low speed for 1 h. The sealed NC membrane was placed into an antibody incubation box and incubated with the primary anti-CBS antibody (purchased from Proteintech (14787-1-AP) overnight on a low-speed shaker at 4 °C. The membrane was washed with TBST three times for 10 min each. The membrane was incubated with a secondary anti-β-acting antibody (prepared with Nakassugi Jinqiao TA-09 1:4000) at room temperature, shaken at low speed

on a shaker for 1 h, and washed three times with TBST again for 10 min each. The images were developed in an exposure apparatus, and the results were observed.

### Functional enrichment analysis
The "org.Hs.eg.db" tool in R package was used to convert genes to gene names. Gene ontology (GO) and Kyoto Encyclopedia of Genes and Genomes (KEGG) data were examined using "clusterProfiler" software (*Wu et al., 2021*).

### Immune infiltrate analysis
CIBERSORT is a method in R software used to calculate immune cell infiltration and estimate the abundance of immune cells. CIBERSORT provides 22 common types of immune-infiltrating cells. We analyzed the association between RSs and tumor immune-invasive cells using CIBERSORT package analysis. The "heatmap" and "ggplot2" packages in R were used for visual drawing.

### Analysis of statistics
Statistical analyses were conducted using RStudio. A $p$-value of $< 0.05$ was considered statistically significant. $*p < 0.05$; $**p < 0.01$; $***p < 0.001$.

## RESULTS
### Identification of differential genes associated with novel modes of death and survival in HCC
We performed a differential analysis of the RNA-seq data of 358 patients with HCC in the processed TCGA database. The volcano plot displayed that 7,776 DEGs were upregulated, and 1,897 DEGs were downregulated (Fig. 1A). To find out ferroptosis genes in DEGs, 259 downloaded ferroptosis genes were intersected with DEGs, and 70 DEGs associated with ferroptosis were screened out (Fig. 1B). Twenty-five downloaded cuproptosis genes were intersected with DEGs to identify cuproptosis genes among the DEGs, which yielded six DEGs associated with cuproptosis (Fig. 1B).

Univariate Cox regression analysis was performed to explore whether the above ferroptosis-related differential genes were related to the survival of liver cancer patients, which identified 31 survival-related ferroptosis genes (SFGs) (Fig. 1B) and three survival-related cuproptosis genes (SCGs) (Fig. 1E). A heatmap was used to show the expression of the 31 SFGs and three SCGs in the samples (Figs. 1D and 1F).

### Development of a predictive survival model
Lasso Cox regression analysis and multivariate Cox regression analysis were performed on the 31 SFGs and three SCGs genes to identify survival-related ferroptosis and cuproptosis genes and construct a survival model (Figs. 2A, 2B). LASSO Cox regression analysis yielded 10 degrees of freedom, aiming to mitigate overfitting and enhance predictive power. Finally, multifactorial Cox regression analysis identified six genes, including SLC1A5, SLC7A11, CBS, GABARAPL1, FLT3, and MT3 (Fig. 2C). Three survival-related cuproptosis genes were identified, including ADM, CDKN2A, and GLS (Fig. 2C). The formula used to calculate the RS was: $RS\_Fe = (0.13046485 \times MT3) + (0.04439969 \times SCL7A11) +$

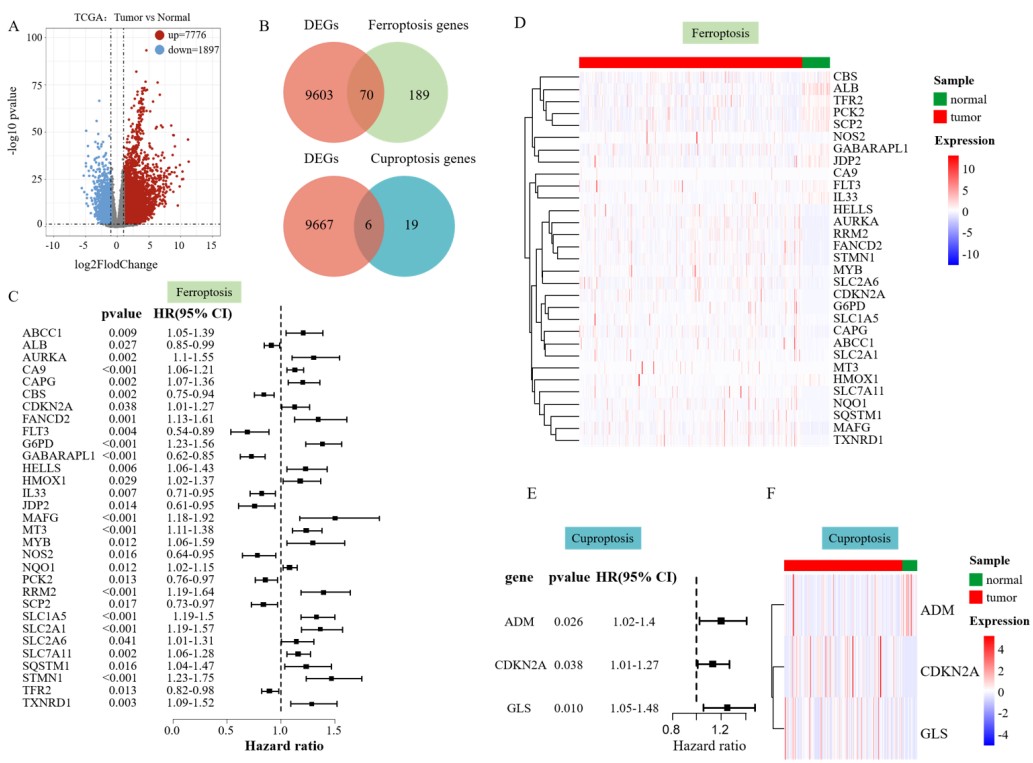

**Figure 1 Differential analysis results.** (A) Volcano plot of DEGs in TCGA data on HCC. (B) The Venn diagram illustrates the intersection of DEGs and ferroptosis genes. (C) Hazard ratio (HR) forest map of ferroptosis genes associated with HCC survival, with HRs > 1 for protective genes and HRs < 1 for risk genes. (D) Heatmap of 31 SFGs. (E) Hazard ratio (HR) forest map of cuproptosis genes associated with HCC survival. (F) Heatmap of three SFGs.

$(0.17832007 \times SLC71A5) + (-0.35683220 \times FLT3) + (-0.15766065 \times GABARAPL1) + (-0.04823521 \times CBS)$, RS_Cu $= (0.1414240 \times ADM) + (0.1036182 \times CDKN2A) + (0.2025281 \times GLS)$.

The patient risk score was performed according to the survival model and its effect was verified. The RS of each patient was evaluated and categorized into low and high RS groups using the median value. The role of the patient RS in the survival model constructed using ferroptosis-related genes was first verified. The heatmap showed an increased expression of CBS, FLT3, and GABARAPL1 in the low RS_Fe group and the increased expression of SLC1A5, SLC7A11, and MT3 in the high RS_Fe group (Fig. 2D), implying that these six genes might act as risk elements impacting the outcome of patients with HCC. Survival analysis using KM revealed a lower survival rate in the high RS_Fe group compared to the low ($p = 0.0001$, Fig. 2E). An ROC curve was drawn to evaluate the survival prediction ability of the model. The area under the curve (AUC) for 3 and 5 years was 0.758 and 0.723, respectively (Fig. 2F), consistent with the results of *He et al. (2023b)*. Figures 2G–2H additionally depicts the reduced lifespan of patients in the high-risk category. These results suggest that the survival model established based on the screened survival-related ferroptosis genes could effectively predict the long-term outcome of HCC patients.

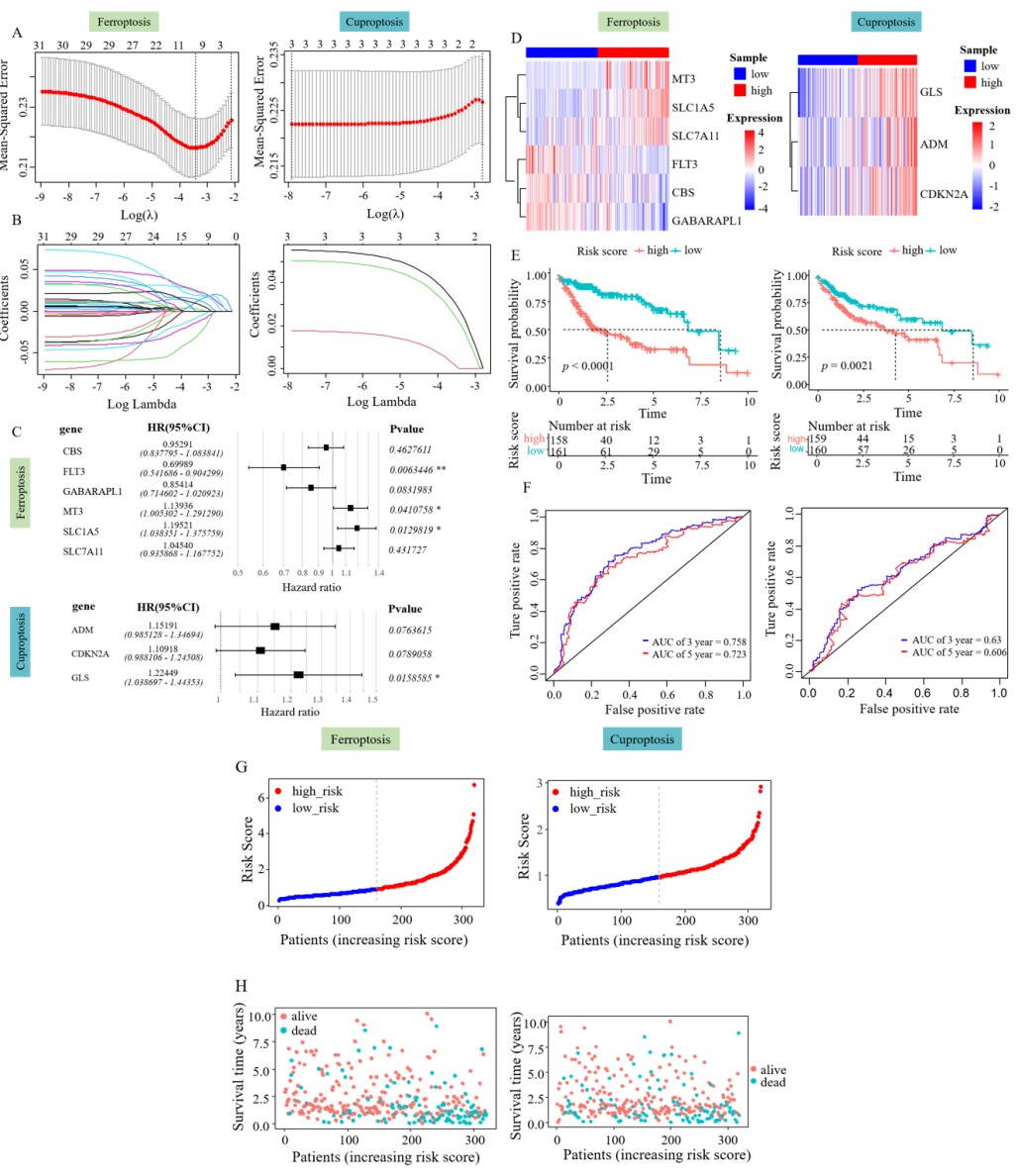

**Figure 2** **Risk assessment model development and evaluation.** (A–B) LASSO Cox regression analysis yielded 10 degrees of freedom. (C) Six SFGs and three SCGs were ultimately pinpointed using multivariate Cox regression. (D) Gene expression heatmap of the two RS groups. (E) KM survival curves in the two RS groups. (F) ROC curves predicting the 3- and 5-year prognosis of HCC patients. (G) Allocation of patient risk assessments in the two RS groups. (H) Survival of high and low RS patients.

Similarly, a survival model based on genes associated with cuproptosis was used for patient risk scoring and validation. The heatmap showed an increased expression of ADM, CDKN2A, and GLS in the high RS_Cu group (Fig. 2D), implying that these three genes might act as risk elements affecting the outcome of patients with HCC. Survival analysis using KM revealed a lower survival rate in the high RS_Cu group compared to the low group ($p = 0.0021$, Fig. 2E). An receiver operating characteristic (ROC) curve was drawn

to evaluate the survival-predicting ability of the model. The area under the curve (AUC) for 3 and 5 years was 0.63 and 0.606, respectively (Fig. 2F). Figures 2G–2H additionally depicts the reduced lifespan of patients in the high-risk category. These results suggest that the survival model established based on the screened survival-related cuproptosis genes could effectively predict the long-term outcome of HCC patients.

## Prognostic genes were validated using external data

The GEO and HCCDB databases were used to externally verify the relationship between the constructed survival model and the prognosis of patients with HCC. The study also aimed to evaluate the accuracy of these risk assessment models using the GEO and HCCDB databases. The heatmap showed an increased expression of CBS, GABARAPL1, and FLT3 in the low RS_Fe group, whereas SLC1A5, SLC7A11, and MT3 were highly expressed in the high RS_Fe group, consistent with the TCGA data (Fig. 3A). Kaplan–Meier analysis revealed a notable decrease in survival rates in the high RS_Fe group compared to the low RS_Fe group ($p = 0.0014$, Fig. 3B). For ROC curves, the AUC values were 0.676 at 3 years and 0.625 at 5 years (Fig. 4C). The GEO data also demonstrated poor survival in the high RS_Fe group of patients, consistent with the TCGA data (Figs. 3D–3E). Similarly, the results of the analysis of HCCDB data were consistent with the GEO and TCGA results. The conclusions verified by GEO and HCCDB data were consistent with those obtained from the TCGA database, indicating that the survival model could effectively predict the prognosis of patients with HCC.

Subsequently, we used the same approach to validate the survival model established by survival-related cuproptosis genes using the GEO database. The heatmap showed an increased expression of ADM, CDKN2A, and GLS in the high RS_Cu group, consistent with the TCGA data (Fig. 3F). Kaplan–Meier analysis revealed no statistically significant difference in survival rates between the two groups ($p = 0.22$, Fig. 3G). GEO data validation found poor results (Figs. 3H–3J). This could be because studies on copper-mediated death are relatively new and not abundant, and the use of cuproptosis genes to construct survival models is even rarer, so cuproptosis may not be suitable for model validation.

## Construction of the nomogram

In order to verify the effect of RSs and other clinicopathological factors on the prognosis of HCC obtained by survival model. The univariate Cox analysis showed a notable correlation between the overall survival (OS) of HCC patients and factors such as RS, tumor stage, distant metastasis (M), and lymph node metastasis (T) ($p < 0.05$, Fig. 4A). In multivariate Cox regression analysis, only M and RS showed a correlation with patient OS (Fig. 4B). The above results suggest that M and RS can be used as factors influencing the prognosis of patients with HCC. Drawing from the TCGA database findings, a predictive nomogram for 1, 3, and 5-year survival rates in HCC patients was created utilizing M and RS (Fig. 4C).

## Prediction of gene expression in the model

GEPIA expression analysis showed that there was no significant difference in the expression of SLC1A5, SLC7A11, GABARAPL1, FLT3, and MT3 between HCC and normal hepatocytes. In contrast, the expression of CBS in normal hepatocytes was different

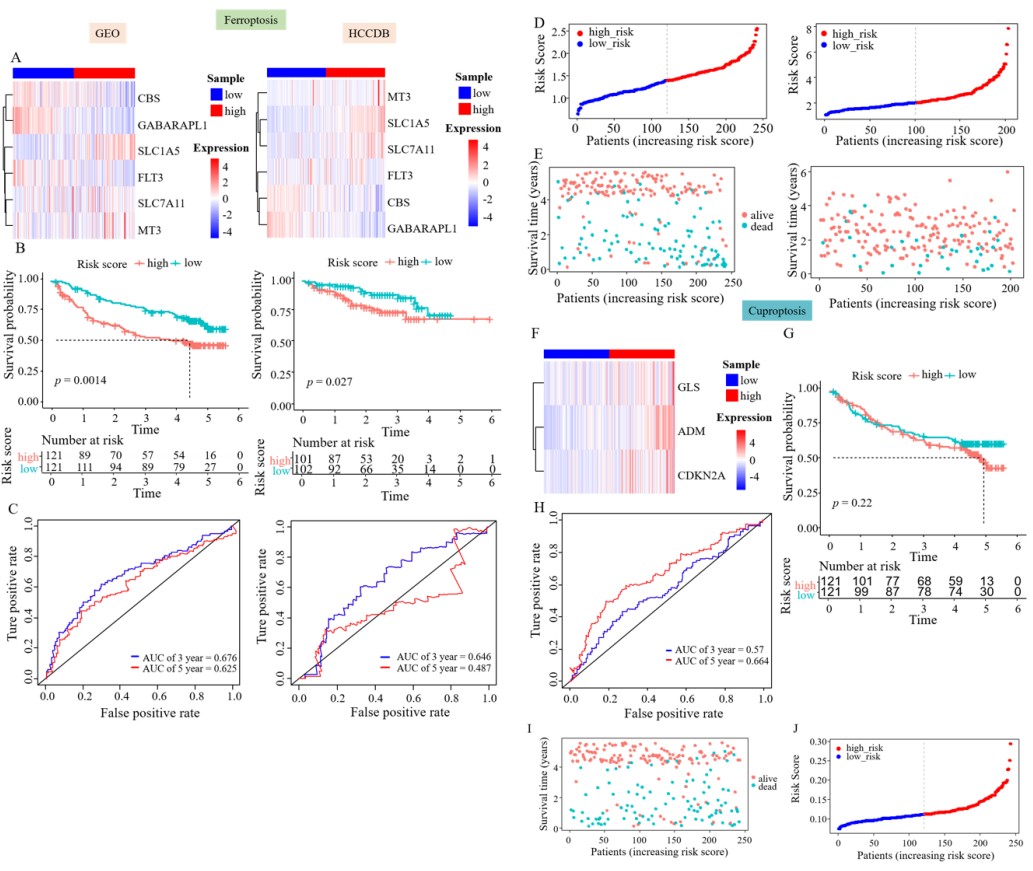

**Figure 3 Prognostic genes were validated using external data.** (A) Gene expression heatmaps of the GEO and HCCDB databases. (B) KM survival curves of the RS_Fe group in the GEO and HCCDB databases. (C) ROC curves of the RS_Fe group predicting 3- and 5-year prognosis in HCC patients. (D) Allocation of RS_Fe group patient risk assessments in the GEO and HCCDB databases. (E) Survival time of RS_Fe group patients in the GEO and HCCDB databases. (F) RS_Cu group gene expression heatmap. (G) KM survival curves of the RS_Cu group in the GEO database. (H) ROC curves of the RS_Cu group predicting 3- and 5-year prognosis in HCC patients. (I) Survival time of RS_Cu group patients in the GEO database. (J) Allocation of RS_Cu group patient risk assessments in the GEO database.

compared to HCC cells (Fig. 5A). Protein validation in the HPA database confirmed the reduced expression of CBS in HCC (Fig. 5B). The qRT-PCR and WB results were consistent with these findings (Figs. 5C–5D).

## Enrichment analysis and immune correlation analysis of prognostic genes

GO analysis of six survival-related ferroptosis genes revealed a collective enrichment of 213 functional enrichment items (Fig. 6A). Subsequently, KEGG enrichment analysis was carried out, revealing eight KEGG enrichment pathways, which predominantly focused on carbon metabolism, autophagy, amino acid metabolism, and other pathways in cancer centers (Fig. 6B). Similarly, we conducted GO and KEGG analyses on the cuproptosis genes (Figs. 6C–6D). We then intersected the GO pathways enriched for survival-related ferroptosis genes with those enriched for survival-related cuproptosis genes and found seven

A

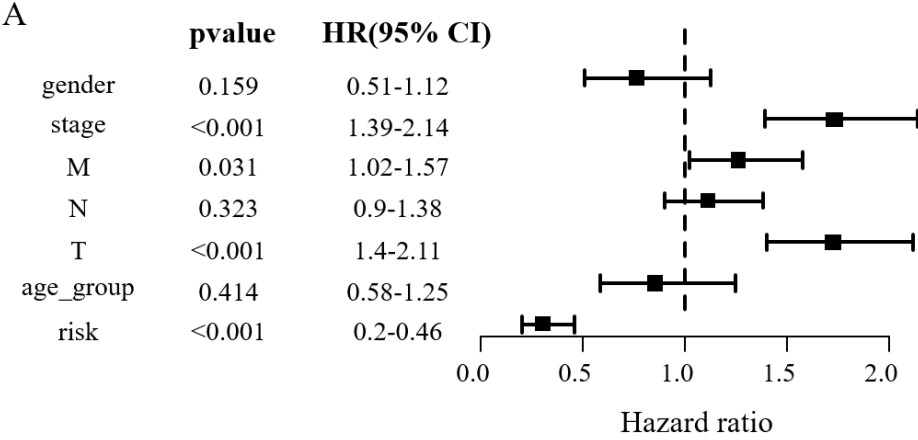

B

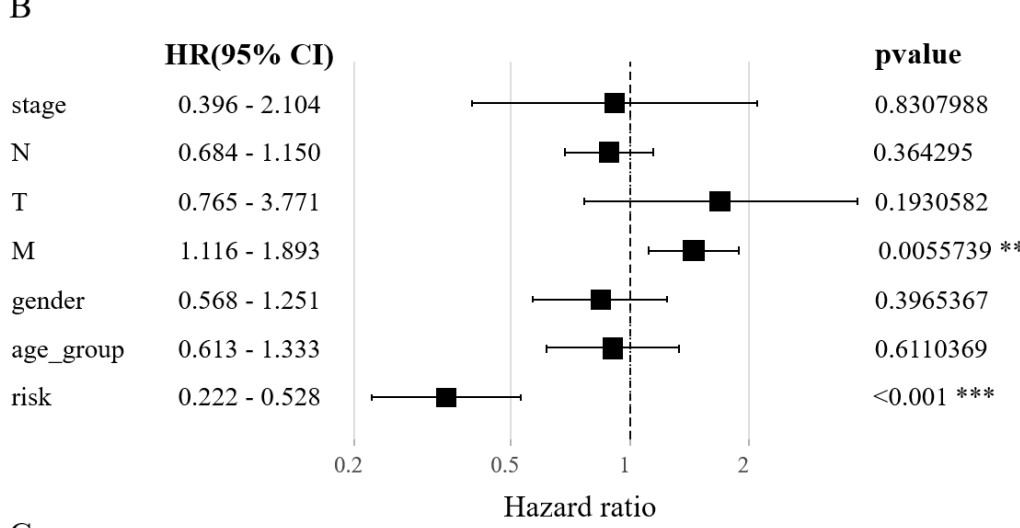

C

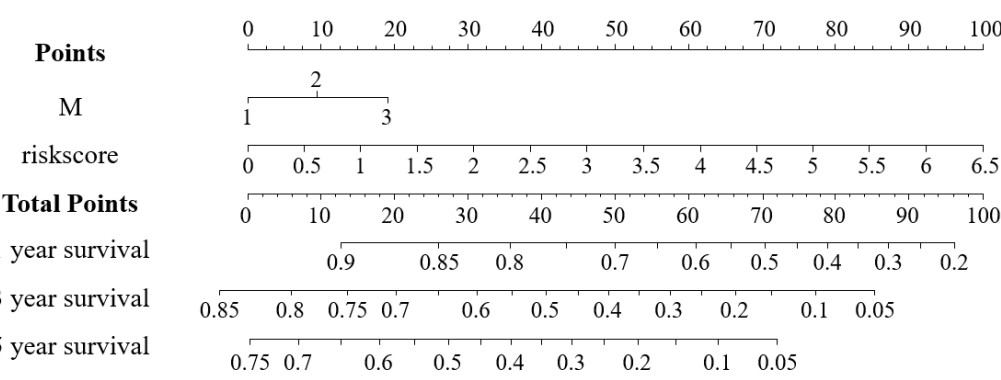

**Figure 4** **Construction of the nomogram.** (A) Clinical elements that affect the prognosis of patients with HCC were identified using univariate Cox regression analysis. (B) Multivariate Cox regression analysis further identified the clinical elements affecting the prognosis of HCC patients. (C) M and RS were used to build nomograms.

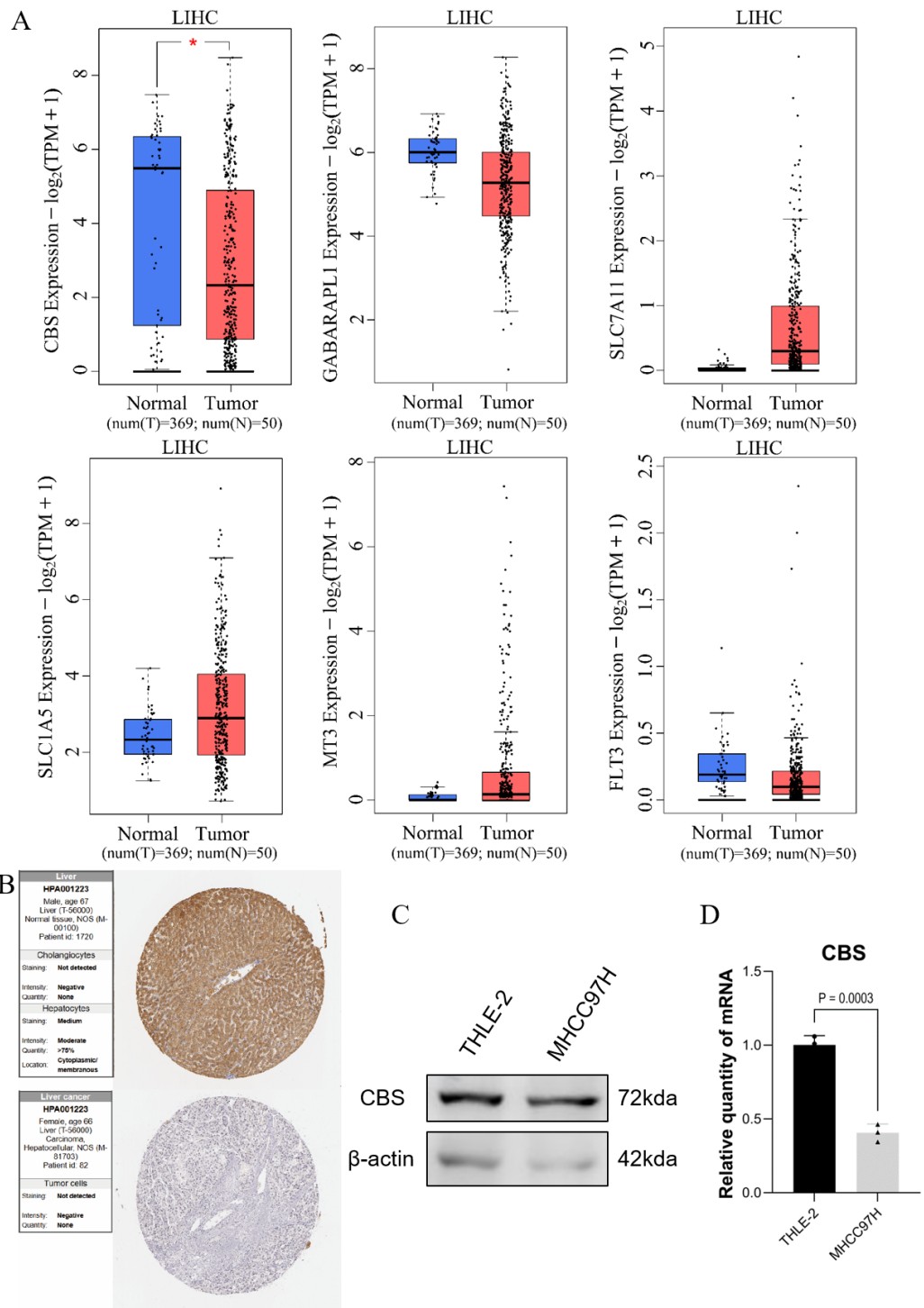

**Figure 5** **Prediction of gene expression by the model.** (A) GEPIA expression analysis. (B) Protein validation in the HPA database. (C) Western blotting of CBS expression levels in THLE-2 and MHCC97H cell lines. (D) qRT-PCR analysis of the expression level of CBS in THLE-2 and MHCC97H cell lines.

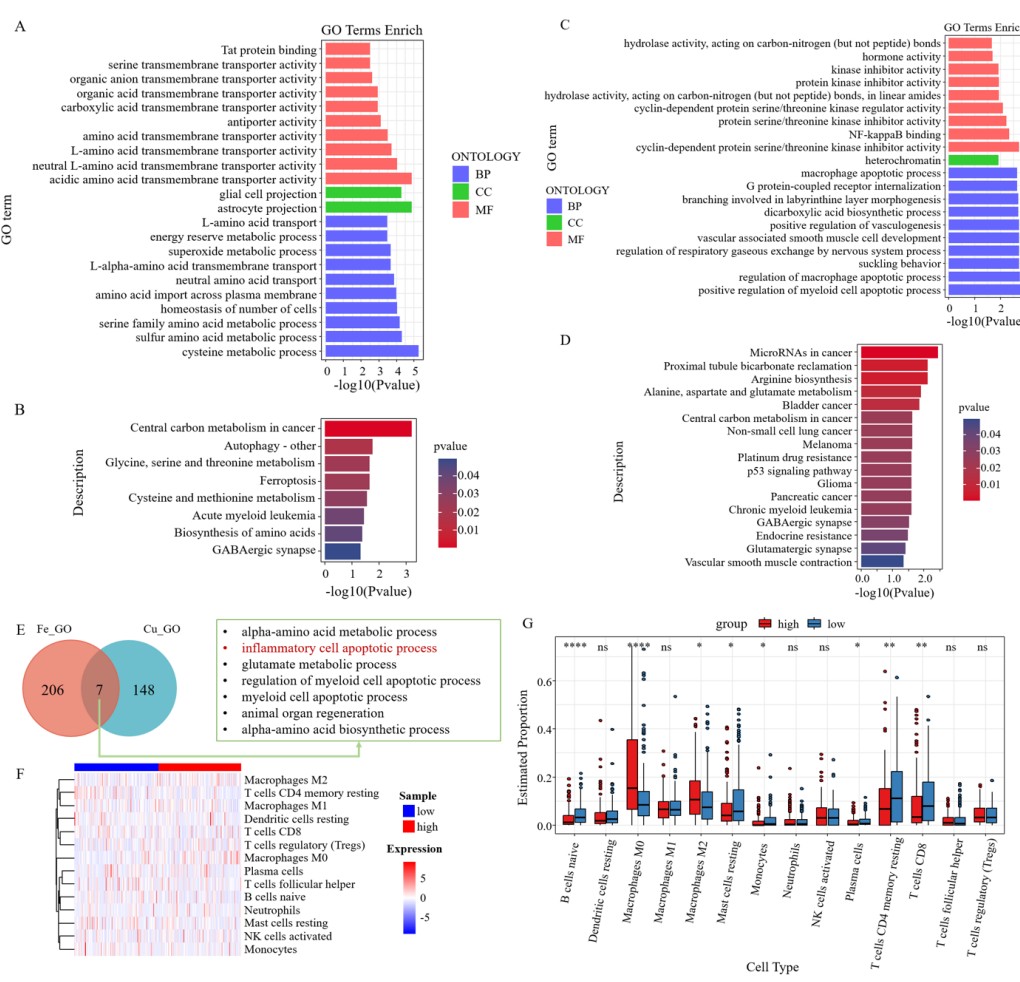

**Figure 6** **Enrichment analysis and immune correlation analysis of prognostic genes.** (A) GO analysis of six survival-related ferroptosis death genes. (B) KEGG analysis of six survival-related ferroptosis death genes. (C) GO analysis of six survival-related cuproptosis death genes. (D) KEGG analysis of six survival-related cuproptosis death genes. (E) GO pathways enriched for survival-related ferroptosis genes intersected with those enriched for survival-related cuproptosis genes. (F) Heatmap of the response of immune cells in the RS_Fe groups based on RSs. (G) Boxplot of immune cells in the RS_Fe groups.

common pathways, including the inflammatory cell apoptotic process and immune-related pathways (Fig. 6E).

Additionally, we found that the enriched pathway was mainly related to immunity. We further analyzed the risk assessment models for tumor immune infiltration. The immune response heatmap is displayed in Fig. 6F. We observed that the high RS group had significantly higher proportions of M0 and M2 macrophages, whereas the low RS group exhibited high proportions of B cells naive, resting mast cells, monocytes, resting CD4+ memory T cells, and CD8+ T cells (Fig. 6G).

## DISCUSSION

Identifying reliable and valid prognostic biomarkers for HCC is critical, and novel death modality-related genes represent promising biomarkers. However, the impact of these genes on the clinical progression of HCC remains poorly understood. Therefore, research into the underlying molecular processes of HCC is necessary to improve patient prognosis through early diagnosis and treatment. This study analyzed public HCC data to obtain survival-related ferroptosis and cuproptosis genes, and the evidence was analyzed and modeled for the screened genes.

This study established a ferroptosis-centric six-gene signature that robustly predicts HCC survival, validated across multiple cohorts, and experimentally confirmed for CBS. SLC7A11, called xCT, constitutes part of the cystine-glutamate reverse transporter protein, also recognized as system Xc (-)). This process improves glutathione production through cystine absorption and glutamate discharge, shields cells from oxidative damage, preserves cellular redox equilibrium and obstructs cell death due to lipid peroxidation (*Chen et al., 2023*; *Chen et al., 2022*). Studies have shown that SLC7A11 has a prevalent expression level in various cancers, and its elevated expression may hinder ROS-triggered ferroptosis, encourage tumor expansion, and foster resistance to medication (*Benedetti, Jézéquel & Orlandi, 1988*; *He et al., 2023a*; *Jiang et al., 2015*; *Park et al., 2010*; *Wang et al., 2017*). Studies indicated that individuals with SLC7A11-positive expression, a key regulator of HCC metastasis, experience reduced OS and increased relapse rates compared to those without SLC7A11 expression (*Huang et al., 2015*). Moreover, SLC7A11-positive expression is correlated with microvascular invasion, reduced differentiation, and elevated tumor nodular metastasis classification (*He et al., 2021*). Research has revealed that lower concentrations of SLC7A11 increase susceptibility to ferroptosis, and its elevated expression impedes the onset of liver damage and cancer (*He et al., 2023a*). The glutamate transporter ASCT2, also recognized as SLC1A5, a neutral amino acid transporter in the SLC1 family, is notably elevated in various tumor types compared to regular tissues (*Hu et al., 2022*). Its upregulation was shown to enhance the proliferation of HCC cells by increasing glutamine uptake (*Li et al., 2023*; *Zhang et al., 2020*). Numerous research studies have reported that elevated SLC1A5 expression in HCC is linked to reduced patient survival rates (*Tambay et al., 2024*). Hypoxia stimulates glutamine transport, angiogenesis, and the upregulation of SLC1A5 expression (*Zhang et al., 2024a*). MT3 is an isoform of metallothionein (MT), which can protect cells from injury by inhibiting multiple oxidative stress pathways (*Si & Lang, 2018*). Initial research indicates that MT3 is predominantly found in the central nervous system; however, it is present in several peripheral organs and various human cancers, such as prostate, lung, breast, urothelial, and esophageal tumors, among others (*Felizola et al., 2014*). However, fewer studies have been conducted on HCC (*Si & Lang, 2018*). CBS, also known as cystathionine γ-lyase, is an enzyme responsible for producing cystathionine through the trans-sulfur route. Its activity is tightly regulated due to its crucial role in antioxidant defense and methylation metabolism (*Kim et al., 2009*). Increasing evidence suggests that CBS downregulation observed in HCC is associated with the poor prognosis of patients with HCC and low CBS expression. The overexpression

of CBS was reported to significantly enhance cell apoptosis *in vitro* and suppress tumor growth *in vivo* (*Pan et al., 2016*; *Zhou et al., 2021*). Research indicates that imatinib mesylate cubes (HA-IM-CBs) can stimulate the CD44-regulated caspase-mediated mitochondrial apoptosis pathway, thereby resisting the activity of HCC (*Lai et al., 2022a*; *Nisha et al., 2022*; *Sun et al., 2019*). FLT3 plays an essential role in blood formation and liver growth, indicating that FLT3 signaling is crucial in liver regeneration (*Aydin et al., 2007*; *Daver et al., 2019*; *Lai et al., 2022b*; *Petersen et al., 2003*). The research reported that the FLT3 gene and protein expression were significantly reduced in patients with HCC compared to healthy liver tissue, which was attributed to decreased FLT3 copy numbers. High FLT3 expression is linked to improved OS in HCC patients treated with sorafenib, although HCC patients with high FLT3 expression or increased copy numbers have a worse prognosis (*Cerami et al., 2012*; *Lai et al., 2022a*). GABARAPL1 (GABA type A receptor-associated protein-like 1, also known as GEC1) is a member of the GABARAP family and was initially recognized as one of the first estrogen-regulated genes (*Fonderflick et al., 2022*; *Le Grand et al., 2014*). It is mainly involved in the transport of selective autophagy receptors and various specific autophagy processes, including mitophagy. Studies have suggested a high expression of FOXO-1, as well as its subsequent targets MAP1LC3B and GABARAPL1, in the hepatic tissues of an HFD/STZ cohort, due to the influence of insulin deficiency and HFD (*Ramadan et al., 2022*). However, there are still relatively few studies in this field.

We identified three copper death-survival correlations using the same method and constructed survival models. However, verification using GEO data showed that the results were inconsistent with those of the TCGA data results. This was because the research on cuproptosis is relatively new and not abundant, and few studies have constructed survival models using cuproptosis genes. Thus, when conducting verification, it was found that cuproptosis is not suitable for constructing models.

Compared to previous prognostic models derived from TCGA datasets, our approach demonstrated enhanced robustness through multi-database validation (GEO and HCCDB) and experimental confirmation. While TCGA has revolutionized cancer biomarker discovery, exemplified by studies identifying SCN3B in glioma (*Liu et al., 2024a*), CDK2 in glioma prognosis (*Liu & Weng, 2022*), AIMP1/CNIH4 in head-neck squamous carcinoma (*Li & Liu, 2022*; *Liu & Li, 2022*), and RAD50 in breast cancer (*Kunwer & Hengrui, 2024*), its limitations must be acknowledged. Recent critiques highlight inherent biases in TCGA data, including batch effects from multi-institutional sequencing protocols, the uneven representation of cancer subtypes, and technical artifacts in bulk transcriptomics that may confound biomarker identification (*Hengrui, Zheng & Panpan, 2024*). These issues are particularly relevant given the proliferation of "batch-generated" TCGA studies utilizing similar bioinformatic pipelines, which risk overfitting and reduced clinical translatability (*Liu et al., 2025*).

We subjected six prognosis-related genes to GO and KEGG analyses to understand the ferroptosis process in HCC induced by these genes. The major pathways enriched in KEGG analysis were all immunologically relevant. Thus, a strong association between immunotherapy and ferroptosis is evident. The induction of iron death by immunotherapy has a more potent anti-tumor effect (*Friedmann Angeli, Krysko & Conrad, 2019*; *Huang et*
*al., 2023*; *Jiang, Stockwell & Conrad, 2021*). The GO pathways enriched by ferroptosis genes and those by cuproptosis genes were jointly enriched in the inflammatory cell apoptotic process pathway. This might indicate that the inflammatory cell apoptotic processes involving ferroptosis and copper death could play a crucial role in certain inflammatory diseases (such as arthritis and enteritis), as well as autoimmune disorders. Comprehending the roles of ferroptosis and cuproptosis in immune cell death may offer novel targets for the treatment of these diseases. We found a correlation between KEGG analysis and immune-associated pathways, followed by using CIBERSORT to calculate RS for immune cell infiltration. The study revealed significant differences in M0 and M2 macrophages, naive B cells, resting mast cells, monocytes, resting CD4$^+$ memory T cells and CD8$^+$ T cells. Some studies have revealed that xCT-induced ferroptosis in macrophages markedly elevated PD-L1 levels in these cells and enhanced the effectiveness of anti-PD-L1 treatments against tumors (*Tang et al., 2022*). Inhibiting tumor cell-intrinsic MELK favors the activation of M1 macrophage polarization, inhibiting M2 macrophage polarization and the stimulation of CD8$^+$ T cell attraction (*Tang et al., 2024*). Tumor-infiltrating CD8+ T cells are generally associated with a better patient prognosis, whereas M2-type macrophages are associated with a worse prognosis. Immune infiltration analysis provides a better understanding of the immune escape capacity of the tumor, thereby assessing the survival expectancy of the patient. In summary, the above results demonstrate the value of applying the risk assessment models in this study, further illustrating the importance of the involved factors. This study opens new outlooks for HCC immunotherapy by demonstrating the possibility of these prognostic genes as ferroptosis treatment goals. The prognostic genes in this study also provide a new target for monitoring based on liquid biopsies. These markers, when combined with emerging technologies, such as ctDNA detection, molecular barcoding, and methylation analysis, can achieve non-invasive risk stratification and treatment response assessment for HCC, promoting precise diagnosis and treatment.

The prognostic genes related to ferroptosis/copper death identified in this study (such as the low expression of CBS) are expected to achieve clinical transformation through liquid biopsy technology. Studies on neuroblastoma have confirmed that the dynamic monitoring of circulating tumor DNA (ctDNA) could assess treatment responses and early recurrence in real time (*Jahangiri, 2024*). The risk genes, such as SLC7A11 and CBS, identified in this study can be detected in the plasma of HCC patients using targeted sequencing, providing dynamic prognostic information beyond static tissue biopsies. The molecular barcoding system developed in research on pancreatic and biliary duct malignancies has significantly improved the detection sensitivity of low-frequency mutations (*Ohyama et al., 2024*). The application of this technology to gene combinations, such as FLT3 mutation and CDKN2A deletion, can enhance the early diagnosis of HCC in high-risk populations. The methylation modification of ferroptosis regulatory factors, such as hypermethylation of the SLC7A11 promoter, can be integrated into this model. Breast cancer studies have shown that methylation features in free DNA have high specificity and are suitable for early HCC screening (*Gonzalez et al., 2024*). In clinical applications, the treatment of HCC patients can be performed using risk stratification, efficacy monitoring, and early intervention. In patients with HBV/HCV cirrhosis of the liver blood samples, using adigital PCR/NGS

validation gene combinations (SLC1A5/SLC7A11/CBS/ADM/CDKN2A) (*Zhang, Ding & Zhou, 2024b*), the changes can be quantified in gene expression in ctRNA during sorafenib/immunotherapy to predict ferroptosis sensitivity; combined with imaging examinations (MRI/LI-RADS), early diagnosis and treatment of HCC related to NAFLD can be achieved.

This study also has certain limitations. For example, our risk assessment model was established using a public database, so it is prone to relying on retrospective data and has constraints on the generalizability to different populations. Therefore, it is necessary to partially verify its clinical application using prospective studies (such as *in vivo* and *in vitro* experiments, *etc.*). Our focus was only on the genes related to ferroptosis and cuproptosis, as well as the expected outcomes of patients with HCC. We did not explore other cell death patterns, such as disulfide bond death.

## CONCLUSIONS

This study integrated multiple databases and experimental verifications to establish a prognostic model for ferroptosis in HCC and found that its predictive efficacy was superior to the traditional TCGA model. Moreover, the association between ferroptosis genes and immune cell infiltration provides a theoretical basis for immunotherapy combined with ferroptosis inducers. In the future, prospective clinical studies based on model genes can be conducted to verify its universality among patients of different races and etiologies (such as hepatitis B/C-related HCC). Moreover, the clinical transformation of genes, such as CBS, for use as liquid biopsy markers can be explored, and non-invasive prognostic monitoring techniques can be developed. Meanwhile, the prognostic genes in this study (such as the high expression of SLC7A11 and low expression of CBS) can be used as biomarkers for the prognostic stratification and treatment sensitivity prediction of HCC patients, facilitating the formulation of personalized treatment plans.

## ACKNOWLEDGEMENTS

I would like to thank Professor Pingkun Zhou for their guidance and assistance.

### Funding

This research was funded by the Natural Science Foundation of China (grant/award number: 82203982), the Beijing Natural Science Foundation (7232107), the Natural Science Foundation of Hunan Province (No. 2021JJ30592), the Health Commission Scientific Research Project of Hunan Province (No. D202309037942). The funders had no role in study design, data collection and analysis, decision to publish, or preparation of the manuscript.

### Grant Disclosures

The following grant information was disclosed by the authors:

Natural Science Foundation of China: 82203982.
Beijing Natural Science Foundation: 7232107.
Natural Science Foundation of Hunan Province: 2021JJ30592.
Health Commission Scientific Research Project of Hunan Province: D202309037942.

## Competing Interests

The authors declare there are no competing interests.

## Author Contributions

- Pengyuan Lei conceived and designed the experiments, performed the experiments, analyzed the data, prepared figures and/or tables, authored or reviewed drafts of the article, and approved the final draft.
- Wenfeng Li performed the experiments, analyzed the data, prepared figures and/or tables, and approved the final draft.
- Dafei Xie conceived and designed the experiments, authored or reviewed drafts of the article, supervision, and approved the final draft.
- Hua Guan conceived and designed the experiments, authored or reviewed drafts of the article, supervision, and approved the final draft.
- Xin Huang conceived and designed the experiments, authored or reviewed drafts of the article, resources, and approved the final draft.
- Bo Huang conceived and designed the experiments, authored or reviewed drafts of the article, supervision, and approved the final draft.

## Data Availability

Raw data is available in the Supplemental Files.

The TCGA-LIHC dataset is available at:

https://xenabrowser.net/datapages/?cohort=GDC%20TCGA%20Liver%20Cancer%20(LIHC)&removeHub=https%3A%2F%2Fxena.treehouse.gi.ucsc.edu%3A443.

Sequence data is available at NCBI: GSE14520.

The HCCDB18 dataset is available at:

http://lifeome.net:809/#/download accession number HCCDB18.

## Supplemental Information

Supplemental information for this article can be found online at http://dx.doi.org/10.7717/peerj.19899#supplemental-information.

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
