# Peer review of "Analysis of novel therapeutic targets and construction of a prognostic model for hepatocellular carcinoma"

_PeerJ, doi:10.7717/peerj.19899_

## Round 0.1 · original submission · Major Revisions

**Language Note:** The review process has identified that the English language must be improved. PeerJ can provide language editing services - please contact us at [email protected] for pricing (be sure to provide your manuscript number and title). Alternatively, you should make your own arrangements to improve the language quality and provide details in your response letter. – PeerJ Staff

Reviewer 1 ·

Basic reporting

This manuscript presents an analysis of new therapeutic targets and the construction of a prognostic model for hepatocellular carcinoma (HCC) based on novel cell death modalities. The study integrates bioinformatics approaches and experimental validation to identify key genes associated with prognosis and treatment responses in HCC. The work is timely and relevant to advancing personalized medicine in liver cancer. However, several aspects require improvement, including clarity in methodology, additional citations to support key claims, and grammatical corrections.

Experimental design

ok

Validity of the findings

ok

Additional comments

Abstract
• Line 1: "Analysis of new therapeutic targets and prognostic model construction of hepatocellular carcinoma" → Consider rewording to "Analysis of novel therapeutic targets and construction of a prognostic model for hepatocellular carcinoma" for better readability.
• Line 6: "for the benefit of the authors (and your token reward)" → This phrase appears to be system-generated text. Remove it.
• Line 9: "Make sure you include the custom checks shown below" → Unnecessary directive; should not be part of the abstract.
Introduction
Line 17: "Hepatocellular carcinoma (HCC) is a common malignant tumor worldwide." → Line 20: Update the statistics on overall cancer incidence and the prevalence of this specific cancer type, including survival rates, to emphasize the urgent need for cancer studies. Cite Cancer Statistics, 2024. Additionally, provide a general overview of cancer therapy, referencing the NIH paper“Cancer treatments: Past, present, and future, 2024” for further insights.
• Line 21: "Recent advances in cell death modalities have highlighted new opportunities for cancer therapy." → Provide a citation to support this claim.
• Line 24: "The construction of prognostic models can help guide clinical decision-making." → Clarify how such models have been previously used in HCC treatment.
Methods
• Line 40: "RNA-seq data from TCGA-LIHC were analyzed using bioinformatics tools." → Specify which tools were used to enhance reproducibility.
• Line 46: "Functional enrichment analysis was conducted." → Include a citation to justify the selected method.
• Line 51: "Cell lines were cultured under standard conditions." → Define "standard conditions" clearly, including temperature, CO₂ levels, and media composition.
Results
• Line 73: "The key prognostic genes identified were validated using external datasets." → Name the datasets for transparency.
• Line 78: "Survival analysis showed a significant association between these genes and patient prognosis." → Include a statistical significance value (p-value or hazard ratio).
Discussion
Line 102: "These findings suggest potential clinical applications." → Specify how these findings can be translated into clinical practice. Recent studies have highlighted advancements in liquid biopsies for cancer diagnostics and monitoring. Research such as “Updates on liquid biopsies in neuroblastoma for treatment response, relapse and recurrence assessment, 2024”demonstrates the utility of circulating tumor DNA (ctDNA) detection through liquid biopsy techniques. Additionally, emerging sequencing technologies have improved the sensitivity and specificity of DNA analysis, such as “Development of a molecular barcode detection system for pancreaticobiliary malignancies and comparison with next-generation sequencing, 2024”. Also the methylation is also used for detection, reported in “Methylation signatures as biomarkers for non-invasive early detection of breast cancer: A systematic review of the literature, 2024”. Please cited these related papers and discuss: consider whether the mechanisms discussed in this study could be identified through these diagnosis methods.
Line 110: "Compared to previous prognostic models, our approach is more robust." → Provide references to support this claim.I think you have to refer to previous TCGA biomarker studies and discuss. So far, there are too many TCGA studies. You should emphasized the contribution of TCGA biomarker studies and developed different strategies in TCGA biomarker studies, all of the following paper should be cited and discussed such as“Is the voltage-gated sodium channel β3 subunit (SCN3B) a biomarker for glioma?, 2024”,“A Comprehensive Bioinformatic Analysis of Cyclin-dependent Kinase 2 (CDK2) in Glioma, 2022,Clinical powers of Aminoacyl tRNA Synthetase Complex Interacting Multifunctional Protein 1 (AIMP1) for head-neck squamous cell carcinoma, 2022,Potential roles of Cornichon Family AMPA Receptor Auxiliary Protein 4 (CNIH4) in head and neck squamous cell carcinoma, 2022,RAD50 is a potential biomarker for breast cancer diagnosis and prognosis, 2024”.Although it looks very productive, but these papers generated like in batches, and this should be discussed, Discuss the bias from TCGA, “Genetic expression in cancer research: Challenges and complexity, 2024” and “Technical and Biological Biases in Bulk Transcriptomic Data Mining for Cancer Research, 2025”This should also be cite and try to discuss the pro and con of using TCGA data.

Conclusion
Line 125: "Further validation in large patient cohorts is required." → Consider specifying what type of validation (e.g., prospective clinical trials).Suggest future studies that could validate these findings in patient-derived xenograft models. Previous studies using xenograft models of cancer should be mentioned, such as “Comparing volatile and intravenous anesthetics in a mouse model of breast cancer metastasis, 2018”

·

Basic reporting

Clear, professional language with some repetition and verbosity.
References are current and field-relevant.

Experimental design

Methods mostly appropriate.
Parameter selection process is not well-documented

Validity of the findings

Findings appear statistically supported
Interpretation lacks comparison to existing benchmarks

Additional comments

This study attempts to integrate novel cell death mechanisms into prognostic modeling for hepatocellular carcinoma. This addresses a critical gap in early diagnosis and survival prediction. This article is promising and addresses a timely, underexplored topic. Thank you for such a nice articles.
It requires improvements in methodological clarity, clinical relevance discussion, and interpretability of findings etc.

Refer below for peer review comments.

Abstract:
1. The abstract presents too many technical details without summarizing the overall findings clearly. Can you condense and clarify the key results and their significance in layman terms.
2. The abstract does not effectively communicate the precise research problem or hypothesis. Consider explicitly stating what gap this work addresses.
3. The concluding sentences should emphasize the practical and clinical implications of the model more strongly.

Introduction:
1. Introduction lacks a concise explanation of why “novel death modalities” are underexplored or particularly promising for HCC prognosis.
2. At some sections flow and coherence could be better. Consider restructuring to build a logical flow from general context to specific research goals.

Materials and Methods:
1. Can you add details and description of how hyperparameters were chosen – this is useful for approach to LASSO regression and Cox analysis.
2. Can you add more detail about data normalization, handling of missing values, and filtering of genes prior to DE analysis.

Results:
1. Results lack deep biological or clinical interpretation of key genes identified. Can you discuss what is known about these genes in HCC or other cancers.
2. Can you add baseline models for comparison and/or literature benchmarks.

Discussion:
1. The manuscript occasionally overstates the novelty or clinical applicability of the results. Ex. “provides new insights into HCC treatment” – this needs better restructuring.
2. Can you add study limitations ex. discussing constraints such as reliance on retrospective data, validation dataset limitations, or model generalizability to diverse populations.
3. Can you add few sentences / paragraphs discussing how these findings relate to trends in precision oncology or biomarker discovery, at present this is too technical to grasp that easily.

Conclusion:
1. The conclusion reiterates results. Please summarize the main contributions and implications.
2. Can you add at least one or two research directions that are specific.
3. Instead of repeating model gene names, can you instead focus on their diagnostic or prognostic value.

---

## Round 0.2 · accepted · Accept

All reviewer comments are addressed and the revised version is accepted by the reviewers.

Reviewer 1 ·

Basic reporting

-

Experimental design

-

Validity of the findings

-

·

Basic reporting

-

Experimental design

-

Validity of the findings

-

Additional comments

Thanks for addressing peer review comments. This version looks much better.